# Immunolocalization of the AT-1R Ang II Receptor in Human Kidney Cancer

**DOI:** 10.3390/biom13081181

**Published:** 2023-07-28

**Authors:** Antonella Muscella, Leonardo Resta, Luca Giulio Cossa, Santo Marsigliante

**Affiliations:** 1Dipartimento di Scienze e Tecnologie Biologiche e Ambientali (Di.S.Te.B.A.), Università del Salento, Via Provinciale per Monteroni, 73100 Lecce, Italysanto.marsigliante@unisalento.it (S.M.); 2Anatomia Patologica, Università di Bari, Piazza Umberto I, 70121 Bari, Italy; leonardo.resta@uniba.it

**Keywords:** angiotensin II, AT-1R, cancer, immunocytochemistry, kidney, ERK1/2

## Abstract

This study aimed to evaluate AT1-R expression in normal and cancerous human kidneys, how these expressions are modified, and AT1-R functionality. AT-1R mRNA expression, determined by real-time PCR, was detected in all samples. AT-1R mRNA increased in well-differentiated cancer (G1, *p* < 0.01) and decreased 2.9-fold in undifferentiated cancer (G4, *p* < 0.001) compared with normal kidney tissues. Immunocytochemistry analysis showed that the AT-1R was expressed in the normal tubular epithelium. The glomerulus was also immunoreactive, and as expected, the smooth muscle cells of the vessel walls also expressed the receptor. A total of 35 out of 42 tumors were AT-1R positive, with the cell tumors showing varying numbers of immunoreactive cells, which were stained in a diffuse cytoplasmic and membranous pattern. Computer-assisted counting of the stained tumor cells showed that the number of AT-1R-positive cells increased in the well-differentiated cancers. The functionality of AT-1R was assessed in primary cultures of kidney epithelial cells obtained from three G3 kidney cancer tissues and corresponding histologically proven non-malignant tissue adjacent to the tumor. Indeed, Ang II stimulated, in a dose-dependent manner, the 24 h proliferation of normal kidney cells and cancer cells in the primary culture and phosphorylated extracellular regulated kinases 1 and 2. In conclusion, Ang II may be involved in the growth or function of neoplastic kidney tissue.

## 1. Introduction

In addition to renin, as part of the circulating renin-angiotensin system (RAS), the kidney contains all the components of the RAS and can form angiotensin II (Ang II) as a tissue or paracrine hormone [1,2], and thus intrarenal compartments contain Ang II in concentrations exceeding those in the systemic circulation [3]. Ang II has multiple effects on renal function, the most notable of which are the modulation of renal blood flow, glomerular filtration rate, tubular epithelial transport, renin release, and cellular growth [4]. In cultured proximal tubular cells, Ang II potentiates epidermal growth factor-related mitogenesis [5] causes hypertrophy and induces oncogene expression [6]. All the known intrarenal effects of Ang II are essentially mediated by the type 1 receptor (AT-1R), while the type 2 receptor plays a role in the development and maturation of the fetal kidney.

Ang II has a crucial role in maintaining renal excretory function under extreme conditions, such as hemorrhagic hypotension, dehydration, and renal artery stenosis [7]. Several animal models suggest that Ang II may also be involved in kidney hypertrophy, the pathogenesis of glomerulosclerosis, and tubulointerstitial fibrosis [6]. These actions of Ang II on the kidney in protective and pathological situations demand detailed knowledge of its cellular effects, including the Ang II receptor profile. Given the ability of Ang II to modulate renal vascular and transport function, knowledge of AT-1R localization is critical in the understanding of the segment-specific functions of intrarenal Ang II. AT-1R has been localized in the human kidney by a variety of techniques [8,9,10].

The introduction of AT-1R monoclonal antibodies [11] has made a specific tool available for the localization of the AT-1R receptor in the normal and neoplastic human larynx [12] and breast [13] and in endometrial adenocarcinoma [14]. An AT-1R antibody has been used here to identify nephron segments and cell types expressing the AT-1R receptor in the normal human kidney and to show how its expression is modified in pathological states. Furthermore, we also show that AT1-R functions in both normal and cancerous kidney cells in primary cultures.

## 2. Materials and Methods

### 2.1. Patients

Forty-two cases of kidney cancer tissues and 40 cases of normal kidney tissues were used in this study. None of the patients, aged from 47 to 76 years (mean: 58 years), underwent radiotherapy or chemotherapy before surgical resection. Clinicopathological characteristics including the histological type, lymph node metastasis, tumor size, tumor location, smoke history, and nuclear grading were evaluated in this study (Table 1). The histopathological slides of the tumors were revised and limited to cases of “clear cell renal cell carcinoma” (WHO 2016) [15]. The grades assigned according to the International Society of Urological Pathology (ISUP) system [16,17] were as follows: 6 were Grade 1, 9 were Grade 2, 13 were Grade 3, and 14 were Grade 4.

### 2.2. Primary Culture of Kidney Epithelial Cells

Three kidney cancer tissues and corresponding histologically proven non-malignant tissues adjacent to the tumor were obtained after surgery. All the tumors were of Grade 3. Portions of the tissue were immediately sent to the histopathology laboratory for histological diagnosis. Other portions were placed into a transport medium and disaggregated immediately as described previously by Valente et al. [18]. The cultured cells exhibited the characteristic features of epithelial cells (i.e., positive immunocytochemical staining for cytokeratin 19). Contamination from fibroblasts was quantified using an anti-vimentin antibody (Sigma, St. Louis, MO, USA). Their expression was lower than 10%. In all experiments, to obtain enough cells to perform a basic growth assay and ERK1/2 activation, and to avoid variations in the characteristic features of the cells, normal and corresponding tumor kidney epithelial cells were exclusively used in the primary culture at passage 2.

### 2.3. Proliferation Assay

Normal and cancerous kidney cells were seeded at a density of 1.3 × 10^3^ cells/well in 96-well plates in RPMI growth medium with 10% FBS and incubated overnight at 37 °C in a humidified environment containing 5% CO_2_ to allow adherence. The medium was changed to an FBS-free growth medium for 18 h to induce quiescence. Cell proliferation was assessed using the spectrophotometric MTT assay [19]. The MTT solution (5 mg/mL) was added to each well and incubated for 4 h at 37 °C. The medium was aspirated, and the purple crystals of formazan which formed were dissolved by the addition of isopropanol/HCl 0.04 N. The absorbances were measured with a spectrophotometer at 570 nm against a blank (cells without the MTT solution). The absorbances of the cells treated were transformed into percentages with respect to the controls.

### 2.4. RNA Isolation from Tissues and Real-Time PCR Assays

After isolation and storage in RNA later (Ambion Europe, Huntingdon, UK), the tissues were processed for RNA and protein extraction using the AllPrep DNA/RNA/Protein Minikit (Qiagen, Milano, Italy) protocol and reagents, according to the manufacturer’s instructions. Oligonucleotides were purchased from Invitrogen (Life Technologies, Monza, Italy). Amplification products were sequenced and identified by alignment with the respective reference sequences. The procedures for real-time PCR were performed as described previously [20]. Briefly, for each extracted RNA, reverse transcriptions were performed on 0.25–1 µg RNA using the Bio-Rad iScript Select cDNA Synthesis kit (Bio-Rad, Segrate, Italy) according to the manufacturer’s instructions and in the presence of random primers. In addition, qPCR was performed using the iQ SYBR Green Supermix protocol (Bio-Rad) with a Rotor-Gene 3000 (Corbett Research, St. Neots, UK) real-time machine. Gene expression relative quantification was assessed as previously reported [21].

### 2.5. SDS Electrophoresis of Ang II-R

The malignant and non-malignant kidney tissues were homogenized using an Ultra-Turrax homogenizer in an SDS lysis buffer (65 mM Tris-HCl, 5% mercaptoethanol, 20% SDS, pH 6.8) containing 1 µg/mL of each of the protease inhibitors: soybean trypsin inhibitor, leupeptin and aprotinin, and 1 mM of phenylmethylsulphonyl fluoride. The homogenates were centrifuged for 15 min at 30,000× *g* and at 4 °C. The supernatants (100 µg protein) were boiled for 3 min and loaded onto SDS-polyacrylamide gels with 12% acrylamide. Electrophoresis was performed using a Bio-Rad Mini-Protean III electrophoresis cell for 1 h at 200 V (constant voltage). A mixture of six pretrained protein markers (Bio-Rad) was used for molecular weight calibration.

### 2.6. Immunoblotting

After SDS fractionation, the proteins were transferred to Hybond-C nitro-cellulose sheets (Amersham, Bucks, UK) with a semi-dry transfer technique for 1 h in an LKB Multiphor II Nova Blot apparatus at 0.8 mA/cm^2^ of nitro-cellulose, using a transfer buffer containing 48 mM TRIS, 39 mM glycine, 0.375% SDS, and 20% methanol. The electrophoretic blots were washed in a PBS buffer containing 10 mM sodium phosphate and 150 mM NaCl at a pH level of 7.5 and incubated overnight at 4 °C with 3% BSA as a blocking agent. The blots were incubated overnight in PBS 1% BSA containing anti-AT1 monoclonal antibody (sc-515884, 1:500, Santa Cruz Biotechnology, Inc., Dallas, TX, USA) or containing anti-ERK1/2 monoclonal antibody (sc-514302, 1:250, Santa Cruz Biotechnology, Inc., Dallas, TX, USA) at 4 °C. The sheets were washed again and incubated with biotinylated goat anti-mouse immunoglobulin (Amersham, Bucks, UK) for 45 min at room temperature and then with streptavidin-horseradish peroxidase-labeled anti-biotin antibody (Amersham, Bucks, UK) for 45 min at room temperature. The blots were washed extensively in PBS containing 1% Tween 20, and the immobilized antigen was detected with enhanced chemiluminescence (ECL) (Amersham, Bucks, UK).

### 2.7. Immunocytochemistry

Sections (5 µm) of formalin-fixed, paraffin-embedded tissues were collected on poly-l-lysine-coated slides and stained with hematoxylin-eosin. Immunohistochemical studies were performed using the biotin-streptavidin technique on consecutive sections. The sections were rehydrated in a xylene or graded alcohol ladder and treated with 0.3% hydrogen peroxidase-methanol to remove endogenous peroxidase activity. The sections were immersed in phosphate-buffered saline (PBS) containing 0.1% BSA (fraction V, Sigma Chemicals, Milan, Italy) for 15 min and probed overnight at 4 °C with monoclonal antibody anti-AT1 receptor (sc-1173, 1:200, Santa Cruz Biotechnology, Inc., Dallas, TX, USA) in RPMI 1640 culture medium (ICN-Flow Ltd., Bucks, UK). The sections were then incubated for 45 min at 25 °C with a biotin-conjugated goat anti-rabbit antibody (Amersham, Bucks, UK) diluted at a 1:400 ratio in PBS. Horseradish peroxidase conjugated with streptavidin (Amersham, Bucks, UK) diluted at a 1:500 ratio was then added for 45 min at 25 °C, and the color was developed with 0.06% diaminobenzidine. Finally, the sections were counterstained for 1 min in Mayer’s hematoxylin and mounted in an aqueous medium. The control sections were processed similarly, using normal rabbit serum or rabbit monoclonal antibodies from the same subclass.

### 2.8. Image Analysis and Tumor Cell Counting

The sections were mounted on a Zeiss Axioskop microscope linked to a JVC KY-F30 color video camera (Tokyo, Japan), and the images obtained were digitalized and processed using Optilab (Graftek, Voisins Le Bretounneux, France). To reduce random noise due to video camera recording and digitization, each image was taken four times, and an integration function was used which assigned to each pixel the pixel’s middle value. Systematic errors in image brightness caused by uneven illumination of the sample were reduced by using a shade image (i.e., an image captured under the same illumination conditions but without the sample) which was subtracted from the images. Briefly, the software converted the filtered images to digital gray values (0–255), and subsequently, these were correlated to the number of pixels. Black and white images (8-bit) extracted from the original 24-bit color image were designated as thresholder images to generate the corresponding binary images containing the so-called region of interest (ROI). Here, a range of brightness values corresponding to the specifically stained regions in the image was chosen, thus defining the pixels within that range as belonging to the foreground (ROI) and referring all the other pixels to the background. The minimum value of the range used for thresholding was chosen by analyzing the histogram (i.e., the quantitative distribution of pixels per gray level value) of the corresponding negative control serial sections. In the case of the AT1-R immunolocalization studies, these were sections that were similarly treated, except that the AT1-R antibody was replaced with a rabbit monoclonal antibody from the same subclass. The thresholder images were used to generate the corresponding binary images which were then used to mask the originals. Masked images were finally used to quantify the gray level values of the immunoreactive regions. As far as counting the number of immunoreactive cells is concerned, the black and white images (8-bit) extracted from the original 24-bit color image were designated as thresholder images to generate the corresponding binary images, which contained the specifically stained cells which were automatically counted. The minimum value of the range used for thresholding was chosen by analyzing the histogram (i.e., the quantitative distribution of pixels per gray level values) of the corresponding negative serial sections (i.e., sections where the anti-AT-1R antibodies were replaced by a rabbit monoclonal antibody from the same subclass).

All the measurements were performed with an ×100 magnification. For each section, four randomly chosen microscopic fields were analyzed. After digitalization, the analyzed field (corresponding to the microscopic field at ×100) was a rectangle of 200 × 300 mm, which was considered the unit area (UA). 

The number of AT1R-positive cells per UA was calculated as the sum of the AT-1R-positive cells found in 10 analyzed fields divided by 10.

### 2.9. Statistical Analysis

Each experiment was repeated at least four times. The data points reported in the figures are given as means ± standard deviation (SD).

The data were analyzed using GraphPad Prism 5.0 Software (GraphPad Software, Dotmatic, Boston, MA, USA). A comparison of the expression values between the normal and tumoral types was performed with a Student’s *t*-test. Statistical analysis was also carried out using ANOVA. When indicated, post hoc tests (Bonferroni–Dunn) were also performed. A *p*-value less than 0.05 was considered to achieve statistical significance.

## 3. Results

### 3.1. AT1-R RNA Expression in Normal and Cancerous Kidney Tissues

The total RNA, isolated from 42 tumoral kidney tissues and 40 normal kidney tissues, was subjected to real-time PCR using the specific primers designed for AT-1R. Thirty-five of the 42 tumoral kidney tissues showed AT-1R mRNA expression.

The AT-1R mRNA levels were higher in the normal tissues compared with the cancer tissues (Figure 1a). When the tumor samples were ranked according to the grading, the AT-1R mRNA increased 1.3-fold in the well-differentiated tumors (G1; *p* < 0.01) while it gradually decreased in the poorly and very poorly differentiated tumors, reaching a 2.9-fold decrement (G4; *p* < 0.01) compared with the normal kidney tissues (Figure 1b).

### 3.2. AT1-R Expression in Normal and Cancerous Kidney Tissues

Examination of the normal and cancerous kidney samples revealed that the antibody detected a protein with a molecular weight of 41 KDa in both the normal and cancer tissues (Figure 2a; Appendix A). Figure 2b shows the normalized densitometric quantification of the AT-1R band in the normal and tumor samples ranked according to the grading.

### 3.3. AT1-R Immunolocalization in Normal and Cancerous Kidney Tissues

Immunohistochemical analysis demonstrated that AT-1R was expressed in both the normal and cancer tissues.

The normal portion of neoplastic kidneys showed positive staining in both the cortical and medullary regions. In the cortex, the receptor was expressed primarily in the proximal tubule cells but also in the glomerular mesangial cells and vascular structures (Figure 3). In addition, Ang II-R was expressed in the distal tubule and collecting duct (Figure 3a). Vascular smooth muscle cell staining was present in the afferent arteriole and larger arteries. In the inner medulla, the collecting ducts, collecting tubules, and epithelial cells of the loop of Henle exhibited specific immunoreactivity. AT-1R was abundant in the endothelium cells of the vasa recta, as expected. AT-1R immunostaining was visualized in a diffuse cytoplasmic and membranous pattern in all positive epithelial and endothelial cells.

The validity of the immunoreaction was confirmed by negative controls immunostained with normal rabbit serum or with rabbit monoclonal antibodies from the same subclass.

Thirty-five of the 42 adjacent cancerous portions of kidney tissue analyzed showed varying numbers of stained cells (Figure 4), and the percentage of positive cells ranged from 21 percent to 80 percent. The remaining seven tumors were AT1 negative. Of the 35 AT1-positive tumors, 14 were G1–2, 11 were G3, and 10 were G4 carcinomas. The staining intensity varied from one tumor to another and from cell to cell, where the immunoreactivity was present in a diffuse cytoplasmic and membranous pattern.

The number of AT-1R-positive cells increased significantly (ANOVA: *p* < 0.0001) in the G1–2 well-differentiated cancers compared with G3 and G4 (Figure 4c; Appendix A).

### 3.4. Association of AT-1R Expression with Clinicopathological Characters in Kidney Cancer

When AT-1R expression was correlated with the patients’ clinicopathological characteristics in kidney cancer, the results showed that AT-1R expression was significantly elevated in the patients with lymph node metastasis in comparison with those without lymph node metastasis (*p* < 0.006; Figure 4d). But no significant association between AT-1R expression and age, tumor size, tumor location, or smoke history was found (*p* > 0.05).

### 3.5. Mitogenic Effects of Ang II and ERK1/2 Activation

The complexity of obtaining a tumoral cell culture from primary tumors prompted us to investigate if cells, after culture manipulation, still retain some tumoral characteristics different from the normal cells. We found that the overall different gene pattern between tumoral and peritumoral-derived cells indicated that there was indeed no cell contamination. All genes were overexpressed in the cancerous compared with the peritumor cells (*p* < 0.05, Student’s *t*-test) (Table 2).

Proliferation was analyzed by a spectrophotometric 3-(4,5-dimethylthiazol-2-yl)-2,5-diphenol-2H-tetrazolium bromide (MTT) assay and confirmed by direct cell counts. Normal and cancerous cancer cells were stimulated with increasing concentrations of Ang II (0, 0.1, 1, 10, and 100 nM) for 24 h in a serum-deprived medium (Figure 5). In the cancer cells, an increase in cell proliferation was already evident at 0.1 nM Ang II, showing the maximal response at 10 nM Ang II (ANOVA: *p* < 0.001) (Figure 5). In the normal cells, Ang II was mitogenic but to a lesser extent, since proliferation was evident at 1 nM Ang II (ANOVA: *p* < 0.001). In both cell types, Ang II-induced cell proliferation (10 nM Ang II) was reduced by pretreatment of the cells for 45 min with 1 µM losartan, a specific inhibitor of the AT1 receptor. Preincubation of the cells with 1 mM CGP4221A, an inhibitor of the AT2 receptor, had no effect (Figure 5). These results indicate that Ang II leads to cell proliferation through the AT1 receptor’s activation. Using the mouse monoclonal antibody anti-pERK, two bands of Mr 41,000 and 42,000 were detected in the cell lysates, corresponding to Tyr-204-phosphorylated ERK 1 and 2, respectively. One hundred nM Ang II was stimulated maximally at 10 min, and the phosphorylation of ERK1/2 started to decrease after 20 min (Figure 6). The time and dose dependence trends were not particularly different between the healthy cells and tumor cells, even if a greater quantitative response in ERK phosphorylation in the tumor cells would seem evident (Figure 6).

## 4. Discussion

Here, we demonstrated the highly prominent AT-1R receptor localization in proximal tubule epithelial cells. In addition, the distal tubules, connecting tubules, cortical and inner medullary collecting ducts, and the renal vasculature were also positive. AT-1R was present on the cell surface membranes and in the cytoplasm of tubular and endothelial cells, according to other studies performed with several tissue types [12,13,22]. This cellular distribution depends on the extent of the receptor occupancy for the hormone, since the receptor may be internalized within the cell and recycled to the cell surface [22,23]. The localization of AT-1R in the proximal tubule, glomerulus, and renal vasculature was expected based on physiological studies in which Ang II enhanced proximal tubule transport and caused microvascular contraction [5] as well as mesangial cell proliferation and contraction [4]. The presence of AT-1R in glomeruli, consistent with previous reports on humans [8,9,10], rabbits [24], and rats [8], was predominantly observed in mesangial cells. In the proximal tubular cells, Ang II modifies sodium, bicarbonate, and volume reabsorption as well as gluconeogenesis [25,26]. Since in nonrenal sites, the majority of Ang II’s effects appear to be mediated by AT-1R [27], and in animal experiments, losartan (DuP753) is an effective antagonist of Ang II-induced renal vasoconstriction [28,29], this suggests that the presence of AT-1R in cortex structures reflects its role in the regulation of renal hemodynamics. The cortical and medullary collecting ducts were AT-1R-positive, in contrast with other studies performed in rats and humans [11,30]. However, there is evidence of the role of Ang II in collecting duct [31,32] and distal tubule [33] transport. The effects of Ang II on acid-base transport in these distal tubular segments were inhibited by the AT-1R receptor blocker losartan, suggesting that the effects are mediated by AT-1R [31,32,33].

The presence of Ang II receptors has been demonstrated in neoplastic tissues such as the liver, adrenal gland, lung, breast, and larynx [22,24,34], as well as in renal cell carcinoma [10]. Given the widespread presence of AT-1R receptors in epithelial tissue and their potential involvement in the growth-promoting activity of Ang II, we investigated the possible relationship between AT-1R and the induction of human kidney tumors. We showed that AT-1R is expressed by cancerous human kidneys, and a much higher percentage of cells was found to express AT-1R receptors in well-differentiated carcinomas (G1–2) compared with poorly (G3) and very poorly differentiated carcinomas (G4). Meanwhile, a significant increase in the AT-1R mRNA and protein levels was measured in the well-differentiated kidney cancer tissues compared with those in the normal kidney tissues. Therefore, it appears that the change in AT-1R expression is an early event in kidney carcinogenesis and may be correlated with the transition from the hyperplastic to the neoplastic condition. These data are no different from those previously obtained by others [35].

Our data also show that AT-1R in neoplastic kidney tissue is functional (i.e., able to bind ligands and generate signaling responses), although in other tissues, the receptor appears to be partly internalized [13,22]. However, as reported in the normal and diseased human breasts and larynx [12,23], both the cancerous and the non-cancerous kidneys expressed a single protein band detected by the anti-AT1-R antibody. Such a protein band, recognized by western blotting analysis of the total kidney protein extracts, had an estimated molecular mass of 41 kDa, which appears to represent glycosylated forms of the AT-1R receptor. Because there was no change in the AT-1R molecular weight in the tumors, it follows that if the functionality is changed, then it arises through variability in abundance or intracellular localization, rather than through further post-translational modification. However, we observed and measured a highly evident biological response in both healthy and tumor cells which was expressed in a clear mitogenic effect. Furthermore, it was also associated with the phosphorylation of the ERK1/2 MAPK. In this respect, we note that the time and dose dependence trends were very similar between the healthy cells and tumor cells.

Altogether, the data demonstrate that in a normal kidney, AT-1R is expressed by renal vascular smooth muscle and is widely distributed along the nephron. AT-1R expression is modified in renal cell carcinoma. This is an important finding, since Ang II is known to be an important factor in the regulation of cell proliferation and cancer cell growth and invasion via AT-1R [36,37,38]. In addition, we noted that AT-1R expression was significantly correlated with lymph node metastasis, which was consistent with studies suggesting that an AII or AT-1R blockade may prevent renal cancer progression by inhibiting angiogenesis [39,40]. Thus, these observations suggested that AT-1R played an important role in the progression of kidney cancer, but further studies are necessary to validate our findings. Since some cancers express renin, ACE, and Ang II receptors, there is the potential involvement of the local renin-angiotensin system in growth-promoting events, therefore raising the prospect that Ang II may be involved in the development of cancer, as suggested by Sun [41]. Finally, the finding that kidney cancer retains AT1R suggests that it may potentially respond by deregulating from its source of angiotensin II.

## Figures and Tables

**Figure 1 biomolecules-13-01181-f001:**
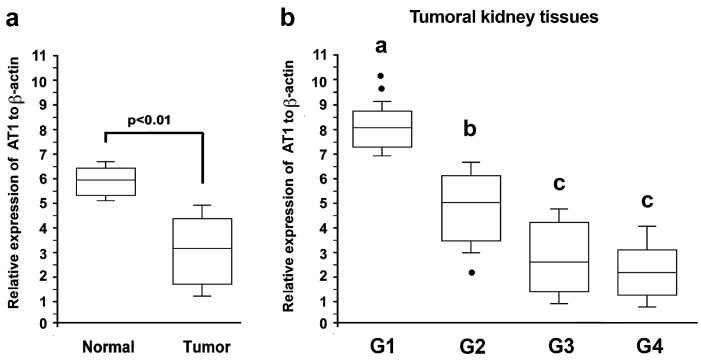
Expression of AT-1R mRNA in the kidney tissue. (**a**) The total RNA was extracted from 42 tumoral kidney tissues and 40 normal kidney tissues, and real-time qPCR analyses targeting the expression of AT-1R were performed (statistical analysis by Mann–Whitney U test: *p* < 0.001). (**b**) Tumoral kidney tissues were classified as G1 (*n* = 6), G2 (*n* = 9), G3 (*n* = 13), and G4 (*n* = 14). Gene expressions were normalized by individual β-actin. Values refer to the mean data ± SD (statistical analysis by ANOVA: *p* < 0.001). Different letters indicate significantly different values under the post hoc Bonferroni test.

**Figure 2 biomolecules-13-01181-f002:**
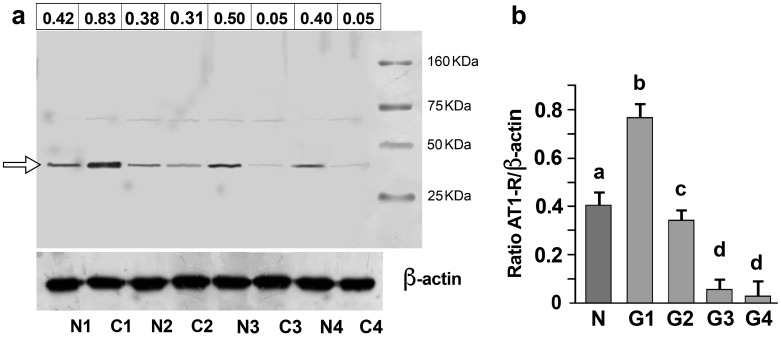
Expression of the AT1-R protein in the kidney tissue. (**a**) SDS analysis, followed by immunoblotting, performed on matched pairs of normal (N1, N2, N3, and N4) and cancerous (C1 (G1), C2 (G2), C3 (G3), and C4 (G4)) kidney samples. In this experiment, the same protein concentration (100 µg) was loaded on the gel. A band of molecular mass of approximately 41 kDa (arrow) was detected. For each sample lane, the densitometric quantification of the AT-1R band refers to the mean data from *n* = 3 different blots. Mean values have been normalized concerning the corresponding β-actin band. (**b**) Densitometric quantification of the AT-1R band obtained by immunoblotting analysis performed on 40 normal (N) and 42 tumoral kidney tissues classified as G1 (*n* = 6), G2 (*n* = 9), G3 (*n* = 13), and G4 (14). Values refer to the mean data ± SD from different blots and have been normalized concerning the corresponding β-actin band. Molecular weight markers are shown on the left (statistical analysis: *p* < 0.001 by one-way ANOVA). Values with shared letters are not significantly different according to Bonferroni–Dunn post hoc tests.

**Figure 3 biomolecules-13-01181-f003:**
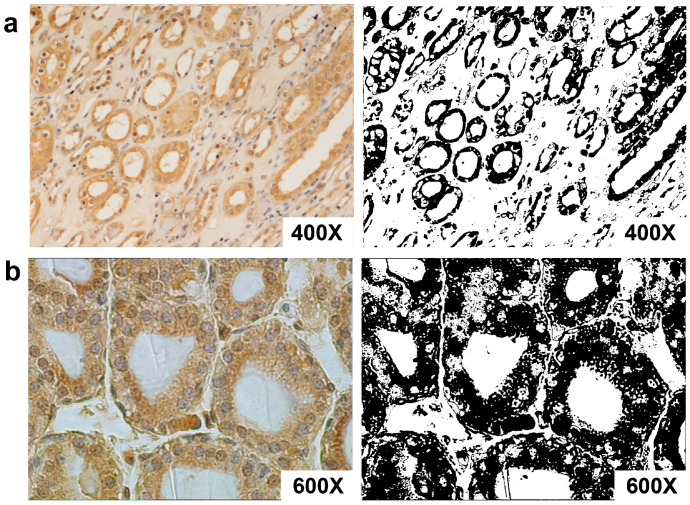
The expression of AT-1R in normal kidney and kidney cancer tissues with immunostaining of paraffin-embedded human kidney. (**a**) AT-1R expression in the kidney at low magnification (400×). AT-1R is expressed in the proximal tubule cells, collecting ducts, and vascular structures. (**b**) AT-1R is expressed in epithelial cells (600×). (Right) Binary images obtained from (**a**,**b**) after thresholding showed the regions stained by the antibody.

**Figure 4 biomolecules-13-01181-f004:**
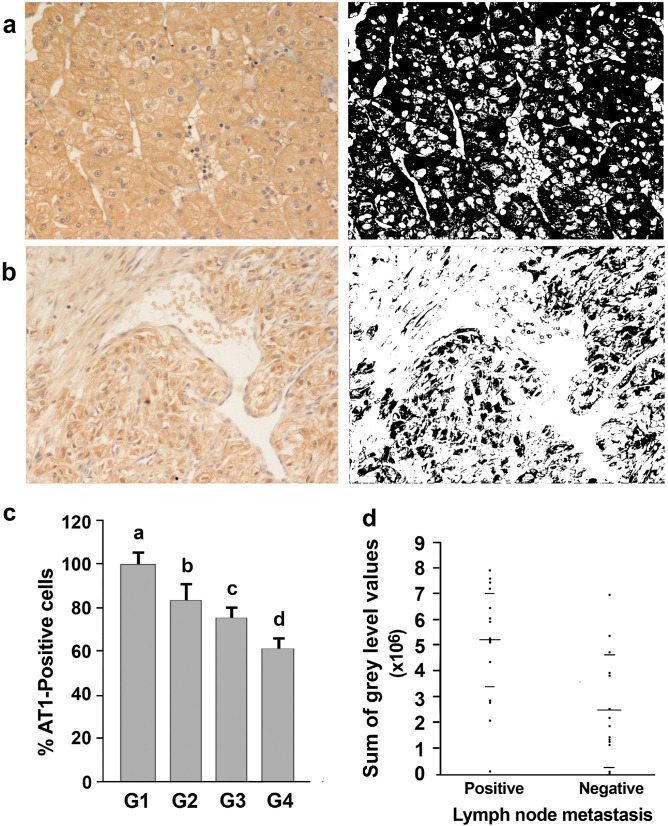
The expression of AT-1R in normal and cancerous kidney tissues. (**a**) The expression of AT-1R in renal cancer G1 (200×). (**b**) Lesser expression of AT-1R in G4 renal cancer both in epithelioid and spindle cells (upper and left portion of the picture (200×)). (Right) Binary images obtained from a and b after thresholding, which show the regions stained by the antibody. (**c**) Distribution of AT-1R-positive cells in the kidney tumors of differing nuclear gradings. Values refer to the mean data ± SD (statistical analysis: *p* < 0.001 by one-way ANOVA). Values with shared letters are not significantly different according to Bonferroni–Dunn post hoc tests. (**d**) Distribution of AT-1R expression values (expressed as the sum of gray levels) into the two groups of patients’ lymph node metastasis statuses. Horizontal long bars represent the mean values, while shorter bars represent standard deviations (Kruskal–Wallis test: *p* < 0.006).

**Figure 5 biomolecules-13-01181-f005:**
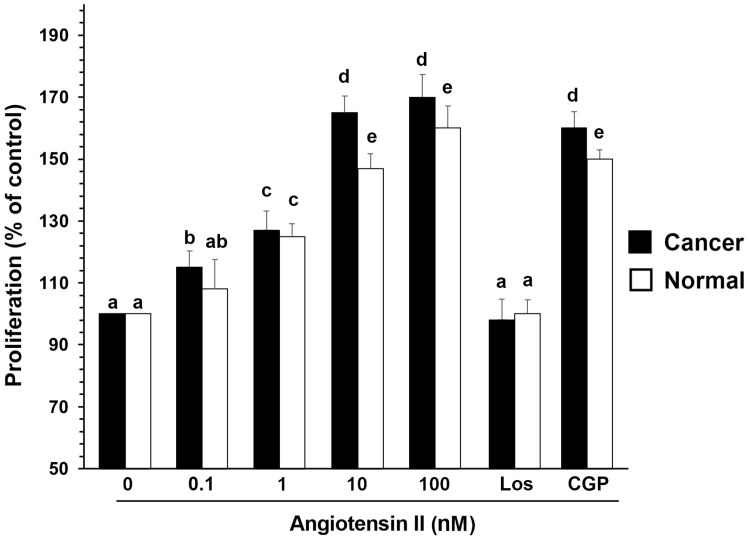
Effects of increasing doses of Ang II upon the proliferation of human cultured normal and cancerous kidney cells and the effects of losartan or CGP4221A. Normal and cancerous kidney cells were stimulated with increasing concentrations of Ang II (0, 0.1, 1, 10, and 100 nM) for 24 h in serum-deprived medium, or the cells were pretreated with 1 µM losartan or 1 mM CGP4221A for 45 min. Proliferation was analyzed by a spectrophotometric 3-(4,5-dimethylthiazol-2-yl)-2,5-diphenol-2H-tetrazolium bromide (MTT) assay. The data are means ± S.D. of five different experiments (*n* = 5). Values with shared letters are not significantly different according to Bonferroni–Dunn post hoc tests.

**Figure 6 biomolecules-13-01181-f006:**
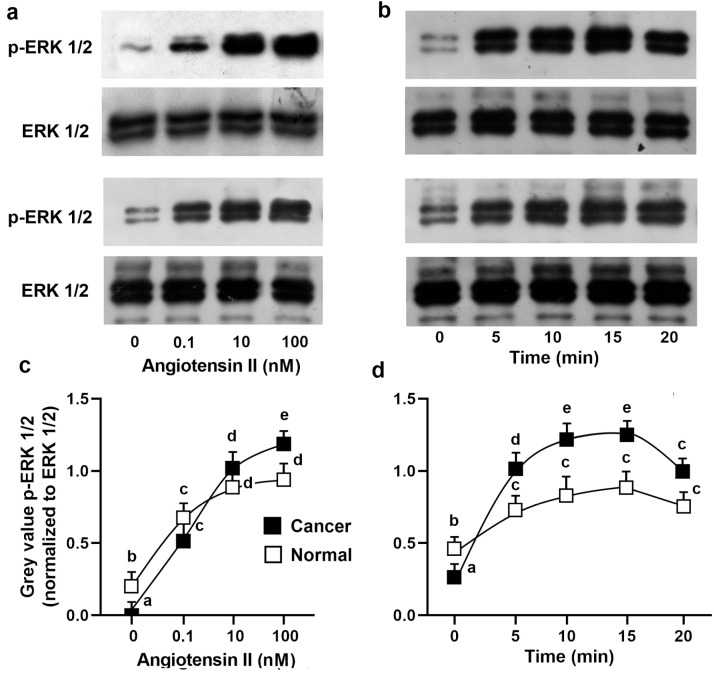
Effects of Ang II on ERK1/2 phosphorylation in human cultured normal and cancerous kidney cells. Normal (up) and cancerous (down) cells were incubated (**a**) without or with 0.1, 10, and 100 nM Ang II for the indicated times or (**b**) with 100 nM Ang II for the indicated time. Lysates were analyzed by western blotting with anti-total-ERK1/2 or with anti-phosphorylated-ERK1/2. (**c**,**d**) Densitometric analysis of phosphorylated-ERK1/2 normalized to unphosphorylated ERK1/2. The data are means ± S.D. of five different experiments (*n* = 5). Values with shared letters are not significantly different according to Bonferroni–Dunn post hoc tests.

**Table 1 biomolecules-13-01181-t001:** Clinicopathological characteristics in kidney cancer.

CLINICOPATHOLOGICAL CHARACTERS	N
Age (years)	
<59	22
≥59	20
Gender	
Male	25
Female	17
Smoke history	
Positive	19
Negative	23
Tumor location	
Left	20
Right	22
Tumor size (cm)	
≤5	27
>5	15
Lymph node metastasis	
Positive	18
Negative	24
Nuclear grading	
G1	6
G2	9
G3	13
G4	14

**Table 2 biomolecules-13-01181-t002:** Description and relative intensity folds of the low-density oligo microarray.

Gene Name	GenBank ID	Function	T/N
*actin*	*NM 001101*	Housekeeping	1.0
*CASP9*	*AB020979*	Apoptosis	1.4
*CCND1*	*M64349*	Cell cycle	3.3
*CCNE2*	*AF106690*	Cell cycle	3.4
*CCNA1*	*NM 003914*	Cell cycle	3.1
*p53*	*NM 000546*	Cell cycle	1.7
*MYC*	*M14206*	Cell cycle	4.0
*CTSD*	*M11233*	Proteolysis	4.7
*c-KIT*	*NM000222*	Tumorigenesis	3.2
*JNKK2*	*AF022805*	Tumorigenesis	3.9
*Survivin*	*AF077350*	Tumorigenesis	9.1

T/N is the ratio of the relative fold (median intensity of each gene/median intensity of actin) in T is the cultured tumoral kidney cells, and N cultured peritumoral kidney cells.

## Data Availability

Not applicable.

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
