# Peer review of "Immunolocalization of the AT-1R Ang II Receptor in Human Kidney Cancer"

_biomolecules, 2023, doi:10.3390/biom13081181_

Round 1

Reviewer 1 Report

The manuscript entitled "Immunolocalization of AT-1R Ang II receptor in Human Kidney Cancer” by Muscella et al. describes the differences of AT1-R expression in human normal kidney and kidney cancers. The authors observed that the number of AT-1R-positive cells increased in well-differentiated tumors and that Angiotensin II stimulated the proliferation of normal and kidney cancer cells. Thus, the authors assume an involvement of Ang II in growth and function of neoplastic kidney tissue, which is not completely new for the community. However, there are only a few studies revealing the role of Ang II in renal cancers and the work presented here is of high interest for the field, even with its relatively low case numbers.

The manuscript is well written and only minor revisions with respect to the English language are needed.

Minor point:

Table 2, Page 7: Please correct the formatting

Author Response

Dear

Reviewer

I would like to thank you for being the referee on our manuscriptImmunolocalization of AT-1R Ang II receptor in Human Kidney Cancer”

We have made appropriate amendments to the manuscript following your suggestions as detailed below.

                                                                       Yours Sincerely

                                                                     Antonella Muscella

The manuscript is well written and only minor revisions with respect to the English language are needed.

Minor point:

Table 2, Page 7: Please correct the formatting

This has been corrected

Reviewer 2 Report

In this manuscript, the authors have made an attempt to investigate the role of AT1-R in renal cancer. They have shown that AT1-R mRNA levels and protein levels were higher in human tumour as compared to normal tissue. In vitro studies suggests Ang II increased the cell proliferation of normal and renal cancer cells in a dose dependent manner.

The manuscript lacks novelty as similar studies have already been reported before by Dolle-Hitz et al (2010) and also Fu et al  (2022) have shown that losartan has been able to reduce tumour in xenograft mouse model derived from renal cancer cells.

In addition, the manuscript lacks detailed molecular mechanism. 

Due to lack of novelty and lack of in depth studies, this manuscript cannot be accepted in the current form.

Author Response

Dear

Reviewer

I would like to thank you for being the referee on our manuscriptImmunolocalization of AT-1R Ang II receptor in Human Kidney Cancer”

We have made appropriate amendments to the manuscript following your suggestions as detailed below. 

                                                                       Yours Sincerely

                                                                     Antonella Muscella

The manuscript lacks novelty as similar studies have already been reported before by Dolle-Hitz et al (2010) and also Fu et al (2022) have shown that losartan has been able to reduce the tumour in xenograft mouse model derived from renal cancer cells.

In addition, the manuscript lacks detailed molecular mechanisms. 

Due to a lack of novelty and lack of in-depth studies, this manuscript cannot be accepted in its current form.

The opinion expressed by the Referee is not shared by us Authors; this communication adds new data which, if nothing else, adds robustness to the published case series. Furthermore, the stratification of the results in the grading is done both with the densitometric results of the westerns and with the analysis of the images of the slides. Finally, we also show cell cultures in only 2 passages used to demonstrate the functionality of the receptor certainly not in healthy tissues, but rather in tumor ones.

Reviewer 3 Report

The authors report a study concerning the immunolocalization of AT1R receptor in Human kidney cancer. They investigated the expression of AT1R in kidney tumors and showed that AT1R expression increase in differentiated cell, while it decreases in undifferentiated cells. Interestingly, the authors also show that angiotensine 2 stimulates both primary culture normal and cancer cells from kidney (from patient with cancer of grade 3) and induced Erk1/2 phrosphorylation. Altogether, the study is well done and sound and the results should be of great interest fro the readership of Biomolecules.

Major points

1.       The authors should indicate in the Materials and Methods section the ethical statement of their study.

2.       Line 316 : The first paragraph of the discussion is an introduction which should not be here. This first paragraph should be placed in the introduction section. The discussion should start at line 322. This first paragraph should finish at line 327 (“... several tissue types [12, 13, 23].”

3.       Concerning the antibody against AT1R, do the authors has a negative control?

4.       RTqPCR : the authors used only one reference gene. Please explain why? Was the expression level of beta-actin similar in normal and tumor samples?

5.       The authors used student’s t test with n=3, please explain. I would have use a Mann and Whitney test instead. I have the same comment concerning the data in table 2  and text lines 252-273.

6.       Figure 2: the westernblot should have molecular weight markers.

7.       Please provide a short explanatory title for each figure.

8.       Figure 3 and 4 : please provide photographs with colors (original 24-bit color images in supplementary materials. The caption of Figure 3 starts with B! Please start with A.

9.       Line 273 : Student’s test with n =?

10.   Please provide a scale bar in all images.

11.   Lines 372-286 : These three paragraphes should be fusionned. I suggest to end this paragraph with an opening and conclusion related to the outcomes of the work without reference.

12.   Could the author propose that their AT1R antibody could be used for clinical investigation ? (cancer diagnostic)

Figure 6 :panels a and b, please add the loading controls.

Panels c : the graph title is cut.

Minor points :

Line 247 > “Kruskall Wallis” should be “Kruskal Wallis”

Line 273 P<0.05

Please indicate the culturing condition (line 66-76)

Line 104 insert a space between number and unit

Author Response

Dear

Reviewer

I would like to thank you for being the referee on our manuscriptImmunolocalization of AT-1R Ang II receptor in Human Kidney Cancer”

We have made appropriate amendments to the manuscript following your suggestions as detailed below.

                                                                       Yours Sincerely

                                                                      Antonella Muscella

The authors report a study concerning the immunolocalization of AT1R receptor in Human kidney cancer. They investigated the expression of AT1R in kidney tumors and showed that AT1R expression increase in differentiated cell, while it decreases in undifferentiated cells. Interestingly, the authors also show that angiotensin 2 stimulates both primary culture normal and cancer cells from the kidney (from a patient with cancer of grade 3) and induced Erk1/2 phosphorylation. Altogether, the study is well done and sound, and the results should be of great interest for the readership of Biomolecules.

Major points

  1. The authors should indicate in the Materials and Methods section the ethical statement of their study.

Done

  1. Line 316 The first paragraph of the discussion is an introduction that should not be here. This first paragraph should be placed in the introduction section. The discussion should start at line 322. This first paragraph should finish at line 327 (“... several tissue types [12, 13, 23].”

We agree with the referee and have moved the first paragraph of the Discussion to the Introduction section

  1. Concerning the antibody against AT1R, do the authors has a negative control?

The specificity of the antibody was controlled in corresponding negative serial sections using mouse monoclonal antibodies from the same subclass.

  1. RTqPCR: the authors used only one reference gene. Please explain why? Was the expression level of beta-actin similar in normal and tumor samples?

For real-time qPCR, we tested 3 reference genes (ACTIN, GAPDH, 18S), and of these, the most stable proved to be that of beta-actin which was therefore used as a reference gene.

  1. The authors used a student’s t-test with n=3, please explain. I would have used a Mann and Whitney test instead. I have the same comment concerning the data in Table 2  and text lines 252-273.

The nature of the data considered is such as to understand them sufficiently borderline in terms of kurtosis and skewness for which the use of both parametric and non-parametric tests is possible. However, following the suggestion of the referee we now indicate the value of P obtained by the Mann–Whitney U test.

  1. Figure 2: the western blot should have molecular weight markers.

This has been done.

  1. Please provide a short explanatory title for each figure.

In fact, figures 4, 5, and 6 did not have a short title, unlike Figures 1, 2, and 3 which did. In the revised version we have also given explanatory titles to figures 4-6.

  1. Figure 3 and 4: please provide photographs with colors (original 24-bit color images in supplementary materials. The caption of Figure 3 starts with B! Please start with A.

This has been done.

  1. Line 273: Student’s test with n =?

Done

  1. Please provide a scale bar in all images.

Done

  1. Lines 372-286: These three paragraphes should be fusionned. I suggest to end this paragraph with an opening and conclusion related to the outcomes of the work without reference.

We agree with the referee in rearranging the indicated paragraphs; this was done in the revised version.

  1. Could the author propose that their AT1R antibody could be used for clinical investigation? (cancer diagnostic)

It could also be employed for this purpose; however, the antibody is commercial and therefore freely available

Figure 6: panels a and b, please add the loading controls.

In these figures total ERK1/2 is shown (ie the phosphorylated form + the non-phosphorylated one) and this represents an excellent loading control.

Panels c: the graph title is cut.

We apologize but, as far as we understand, panel c of Figure 6 shows no cuts of any kind

Minor points:

Line 247> “Kruskall Wallis” should be “Kruskal Wallis”

It has been corrected

Line 273 P<0.05

It has been corrected

Please indicate the culturing condition (lines 66-76)

As already reported in the Methods section, the cultures were carried out following what was described in the work of Valente: PLoS One. 2011; 6(5), e19337. doi 10.1371/journal.pone.0019337.

Line 104 insert a space between the number and unit

It has been corrected

Round 2

Reviewer 3 Report

The manuscript is now suitable for publication. Thanks for answering my questions.  

Author Response

Thanks